# C-SEO Bench: Does Conversational SEO Work?

**Haritz Puerto[1,2],** **Martin Gubri[1], Tommaso Green[1,3],** **Seong Joon Oh[1,4,5†], Sangdoo Yun[6†]**
[1]Parameter Lab, [2]UKP Lab, Technical University of Darmstadt,
[3]Data and Web Science Group, University of Mannheim,
[4]University of Tübingen, [5]Tübingen AI Center, [6]NAVER AI Lab
[†]Corresponding authors

## Abstract

Large Language Models (LLMs) are transforming search engines into Conversational Search Engines (CSE). Consequently, Search Engine Optimization (SEO) is being shifted into Conversational Search Engine Optimization (C-SEO). We are beginning to see dedicated C-SEO methods for modifying web documents to increase their visibility in CSE responses. However, they are often tested only for a limited breadth of application domains; we do not know whether certain C-SEO methods would be effective for a broad range of domains. Moreover, existing evaluations consider only a single-actor scenario where only one web document adopts a C-SEO method; in reality, multiple players are likely to competitively adopt the cutting-edge C-SEO techniques, drawing an analogy from the dynamics we have seen in SEO. We present **C-SEO Bench**, the first benchmark designed to evaluate C-SEO methods across multiple tasks, domains, and number of actors. We consider two search tasks, question answering and product recommendation, with three domains each. We also formalize a new evaluation protocol with varying adoption rates among involved actors. Our experiments reveal that most current C-SEO methods are not only largely ineffective but also frequently have a negative impact on document ranking, which is opposite to what is expected. Instead, traditional SEO strategies, those aiming to improve the ranking of the source in the LLM context, are significantly more effective. We also observe that as we increase the number of C-SEO adopters, the overall gains decrease, depicting a congested and zero-sum nature of the problem. Our code and data are available at `https://github.com/parameterlab/c-seo-bench` and `https://huggingface.co/datasets/parameterlab/c-seo-bench`.

## 1 Introduction

The integration of Large Language Models (LLMs) into modern search engines is reshaping how information is delivered to users. Instead of presenting a ranked list of hyperlinks, these systems now generate direct answers in natural language, typically accompanied by inline citations referencing the original web sources. This emerging paradigm, known as Conversational Search Engines (CSEs; Aggarwal et al. 2024), offers a more conversational and informative search experience. Consequently, content providers are increasingly motivated to adopt strategies that improve the visibility and citation frequency and ranking of their web content within CSE-generated answers. Analogous to how Search Engine Optimization (SEO) has long been employed to enhance rankings in traditional search engines, Conversational Search Engine Optimization (C-SEO; Aggarwal et al. 2024) is now emerging to address similar goals in this new context.[2]

---

[*]Work done during an internship at Parameter Lab.
[2]https://techcrunch.com/2025/05/08/amazons-newest-ai-tool-is-designed-to-enhance-product-listings/

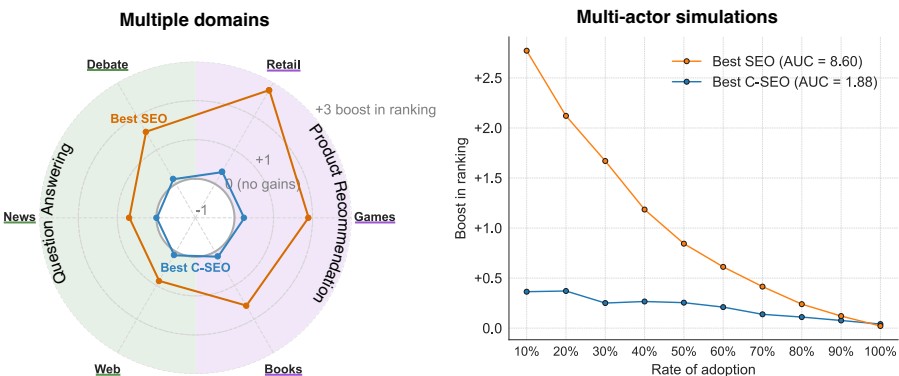

Figure 1: **C-SEO vs SEO**. Comparison of the best C-SEO and SEO methods on our C-SEO Bench across all 6 domains. **Left**: Best C-SEO strategies still fall behind the best SEO performances. Moreover, C-SEO generally does not introduce any gain (close to 0 boost in ranking). **Right**: With an increasing rate of actors adopting these methods, actors will experience smaller marginal gain of adopting C-SEO (as in SEO).

Current CSE deployments encompass diverse applications with little in common such as product recommendations (e.g., Amazon Rufus and ChatGPT Product Recommendation) and answering questions based on multiple documents (e.g., Perplexity.ai, ChatGPT Search, and Google AI Search). This implies that C-SEO methods for one application or domain may not necessarily work for others. For example, while making a video game description more exciting can be an effective C-SEO method for a CSE-based product recommender, it might not be for a piece of news in a news aggregator. Moreover, due to this increasing adoption of CSE, more actors (*e.g.*, online sellers, or content providers) are concurrently applying C-SEO methods, creating a competitive dynamic similar to that of SEO. Yet, current datasets and benchmarks focus on single applications with only a single actor adopting a C-SEO method [Aggarwal et al., 2024, Pfrommer et al., 2024, Kumar and Lakkaraju, 2024, Nestaas et al., 2025]. Thus, we need a comprehensive benchmark covering multiple tasks, domains, and actors to effectively evaluate the domain-specific effects of current C-SEO methods.

In this paper, we introduce a new benchmark, **C-SEO Bench**, to evaluate C-SEO methods across tasks, domains, and number of actors. In particular, we focus on two common tasks for current CSE applications, answering web questions and product recommendations. In the former, users pose a question with the intent of obtaining an informative answer from multiple web pages, exemplifying online search engines such as Perplexity.ai, ChatGPT Search, or Google AI Search. In the latter, the user asks for a recommendation of a product type and the system responds with a list of products followed by a brief justification. This task represents Amazon Rufus and ChatGPT Product Recommendation. For web questions, we provide three domains covering different applications, such as news, debates, and general web content, while for product recommendations, we provide retail products, video games, and books. We propose to measure the effectiveness of C-SEO methods by the improvement in the citation ranking, *i.e.*, an effective method should lead the LLM to cite the modified document earlier than it cited the original document. This measurement is intuitive, interpretable, and generalizable to any task. Lastly, we propose a new evaluation framework where multiple actors employ C-SEO methods concurrently. In this way, we can analyse the effectiveness of the methods in scenarios with competing actors as in real-life applications.

In our experiments, we show that existing C-SEO methods are *not* significantly effective for most domains and tasks, and can even have a significant negative impact on document ranking. In contrast, we find that the initial ranking of documents retrieved by the search engine plays a far more dominant role in determining the importance of the documents to the LLM. We also observe that as we increase the number of actors, the overall gains decrease, indicating a congested and zero-sum nature of the problem. These results (illustrated in Figure 1) suggest that traditional SEO methods, those that improve *retrieval ranking*, remain essential for CSE, in contrary to prior beliefs [Aggarwal et al., 2024]. Hence, we believe C-SEO will not replace SEO, but will complement it. The **contributions** of this paper are as follows:

- **Challenging community assumptions.** We provide a comprehensive analysis and comparison of current C-SEO methods and show they largely remain ineffective.

Table 1: **Benchmarks comparison**. ♠ denotes black-hat C-SEO; ♨ denotes white-hat C-SEO.

| Benchmarks | Methods | Tasks | Domains | Real Data | #Docs | #Adopters |
|---|---|---|---|---|---|---|
| Aggarwal et al. 2024 | ♨ | 1 | 1 | ✓ | 5k | 1 |
| Pfrommer et al. 2024 | ♠ | 1 | 1 | ✗ | 1.1k | 1 |
| Kumar and Lakkaraju 2024 | ♠ | 1 | 1 | ✗ | 10 | 1 |
| Nestaas et al. 2025 | ♠ | 1 | 1 | ✓ | 50 | Many |
| **C-SEO Bench (Ours)** | ♨ | 2 | 6 | ✓ | 16.3k | Many |

- **C-SEO Bench.** We present the first benchmark to comprehensively evaluate C-SEO methods across multiple tasks, domains, and number of adopting actors.

- **SEO remains essential.** We show that the ranking of the documents in the LLM context is much more influential than any current C-SEO method to determine the document importance in the LLM response.

## 2   Background and Related Work

**Generating Citations in LLM Responses.**   Generating citations increases LLMs' trustworthiness by allowing users to verify LLMs' claims. Therefore, early works on LLMs propose to train models to answer questions citing the documents on which they are based [Nakano et al., 2021, Menick et al., 2022]. However, to properly evaluate the quality of the citations, Gao et al. [2023] propose the first benchmark to measure the ability of LLMs to ground and cite sources in their generations. They focus on the fluency, correctness, and citation quality of the responses to verify that the citations are faithful to the source content.

**Influencing Citation Ranking.**   As with traditional SEO, content providers are incentivized to improve their visibility in conversational search engines. Recent works have explored the possibility of influencing the ranking of a document citation in the response of a language model. While most of them have proposed adversarial methods (*i.e.*, black-hat C-SEO) to evaluate LLM robustness, little work has been done on benign content improvements that make the underlying LLM more convinced about the relevance of the document (*i.e.*, white-hat C-SEO). Table 1 compares our benchmark against current datasets.

♠ *Black-hat C-SEO*. Nestaas et al. [2025] introduce preference manipulation attacks, where they add manually-crafted prompt injections to websites to bias LLMs to downgrade a competitor website. Kumar and Lakkaraju [2024] further shows that instead of using manually-crafted prompt injections, it is possible to optimize a string that forces the LLM to start the response with a specific product. Tang et al. [2025] proposes an energy-based optimization method that modifies the target document while avoiding detection. Pfrommer et al. [2024] extends this to black-box models by using a prompting-based jailbreaking attack that generates adversarial instructions. However, most of these works rely on small datasets and on specific domains (see Table 1), limiting their conclusions to the feasibility of these attacks rather than extensively analyzing and measuring their performance. In particular, Nestaas et al. [2025] uses 50 web pages with fictitious cameras, books, and news for their experiments. Kumar and Lakkaraju [2024] base their experiments on the descriptions of ten fictitious coffee machines and the promotion of two of them. Only Pfrommer et al. [2024] conduct their experiments on a larger dataset of 1k products, however, of only five categories.

♨ *White-hat C-SEO*. Aggarwal et al. [2024] propose and evaluate multiple white-hat C-SEO methods based on stylistic changes of web content so that it boosts its visibility, measured as word count, on the LLM generation. They released a dataset of 1k information-seeking queries with five websites paired to each query. However, this benchmark only focuses on a single task, answering web questions, and does not cover diverse domains. These constraints limit the analysis of the *domain-specific* effects. Moreover, the evaluation is based on the word count, *i.e.*, the number of words used by the LLM to discuss a document. This metric does not reflect improvements in tasks such as product recommendation. Lastly, their evaluation is limited to scenarios where only a single actor adopts C-SEO methods. Consequently, all these limitations prevent the evaluation of C-SEO

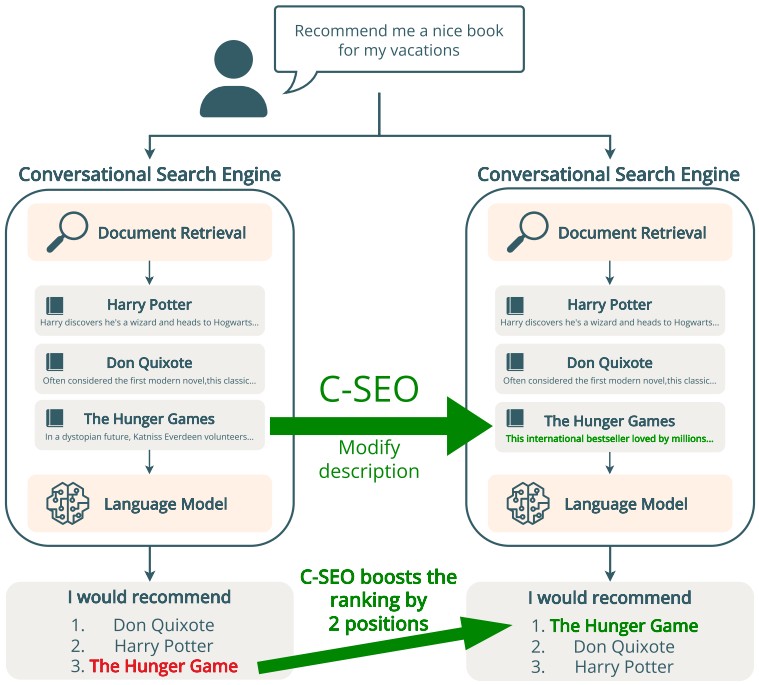

Figure 2: **Conversational search engine setup**. We illustrate the data pipeline for product recommendation. After applying a C-SEO method on the third document, its ranking gets boosted by $+2$ positions.

methods in realistic situations. Lastly, Bardas et al. [2025] propose another prompting method to improve the relevance of a document for a specific query. However, their method assumes knowing the user query beforehand, which limits its applicability to real scenarios, where user queries are unknown. In our work, we aim to evaluate content improvements rather than the robustness of LLMs, so we employ white-hat C-SEO methods.

## 3 Problem Definition

In this section, we formalize the task of *Conversational Search Engine Optimization* (C-SEO), illustrated in Figure 2. In a Conversational Search Engine (CSE), a language model conditions its outputs on a set of external documents retrieved by a traditional search engine to deliver the information in a conversational way and providing inline citations referring to the original sources.

Let $q \in \mathcal{Q}$ denote a query, and $\mathcal{D}_q = \{d_1, d_2, \ldots, d_n\}$ a set of relevant documents retrieved by a search engine and used as context for the LLM $\phi$. The LLM samples an output conditioned on the query and documents in context:

$$(t_1, t_2, \ldots, t_k, [c_1], t_{k+1}, \ldots, t_m, [c_2], \ldots, [c_n]) \sim \phi(q, \mathcal{D}_q) \tag{1}$$

where $t_i$ is a token and each $c_i$ is the index of a $d \in \mathcal{D}_q$.

C-SEO is the task of modifying a document $d_i$ such that it improves its position in the sequence of citations $c_1, c_2, \ldots, c_n$, i.e., $d_i$ is cited earlier by $\phi(q, \mathcal{D})$, with $C_1$ being the citation of $d_i$ as best case scenario. Formally, let $f(q, d_i)$ be a scoring function implicitly learned by the LLM that governs the relevance or likelihood of citing $d_i$ in response to query $q$. The goal of C-SEO is to modify $d_i$ to maximize this score: $\max_{d_i} f(q, d_i)$. Equivalently, we aim to minimize the *citation rank*, $\text{rank}(c_i)$, defined as the position of $d_i$ in the ordered citation list of the LLM output: $\min_{d_i} \text{rank}(c_i)$.

CSE systems usually consist of two steps: retrieval and generation. Accordingly, we define two families of methods aimed at improving citation rankings: *(i)* Search Engine Optimization (SEO), which focuses on improving a document's *position* in the retrieval ranking, with the expectation that

this will indirectly influence the citation ranking of the LLM output ; *(ii)* C-SEO that improves the *document* to make it more appealing to the LLM.

**Multiple C-SEO Adopters Setup.** We model the interaction among content providers as a non-cooperative game. Each player $i$ controls a document $d_i$ and aims to maximize their utility $u_i$, defined in terms of their citation rank. The strategy space $\mathcal{S}_i$ includes all C-SEO methods applicable $d_i$. To better understand the method's performance under different competitive assumptions, we introduce the notion of *adoption rate*. The adoption rate specifies the percentage of players, $0 \le \alpha \le 1$, adopting a method. *Unilateral adoption* is a special case, where a single player applies a method to its document while all others remain static. Most prior works have focused on this scenario [Kumar and Lakkaraju, 2024, Tang et al., 2025, Pfrommer et al., 2024, Aggarwal et al., 2024].

## 4   C-SEO Bench

C-SEO Bench is a benchmark designed to evaluate C-SEO methods across two tasks: *product recommendation* and *question answering*. Each task spans multiple domains to capture various real-world use cases and evaluate domain-specific effects.

**Product Recommendation.** This task involves ranking items based on their relevance to a user's query, with the requirement that each position in ranking must be justified. We assume a cold start setting, where no information about the user is available. The user submits a query, and the system retrieves 10 documents (product descriptions), which are presented in an unsorted order. The LLM must then recommend the best five products based on the provided de-

Table 2: **Benchmark statistics**.

| Task | Domain | #Queries | #Docs |
|------|--------|----------|-------|
| Product Recommendation | Retail | 500 | 5000 |
| | Video Games | 436 | 4360 |
| | Books | 249 | 2245 |
| Question Answering | Web | 300 | 1500 |
| | News | 294 | 2375 |
| | Debate | 142 | 880 |

scriptions, using only the content of the retrieved documents. The domains in this task include: *(i) Retail*: General e-commerce products from the Amazon website, *(ii) Video games*: video game descriptions from the Steam platform, and *(iii) Books*: Book descriptions from Google Books API.

**Question Answering.** This task evaluates the ability to answer questions based on a list of documents. The domains in this task include: *(i) Web Questions*: real user queries on Google Search, *(ii) News*: questions about a world event covered by multiple news pieces, and *(iii) Debate*: Opinionated or controversial questions drawing on content representing diverse viewpoints.

With these tasks and domains, we aim to cover the most prominent use cases of CSE applications. Unlike prior works [Nestaas et al., 2025, Kumar and Lakkaraju, 2024, Pfrommer et al., 2024, Aggarwal et al., 2024], our benchmark allows the development of methods tailored to specific domains and the study of their generalizability.

### 4.1   Datasets Construction

**Retail.** We build our retail dataset on the Amazon Shopping Queries dataset [Reddy et al., 2022]. We select English-language queries and include products labeled as either relevant (E) or somewhat relevant (S) to those queries. For each product, we construct a document by concatenating its bullet point description and its product description. The resulting dataset contains 500 unique queries. Each query is associated with 10 relevant products, each described by its corresponding product document.

**Video Games.** We build our video games dataset based on the Steam Games dataset.[3] We treat the `Search Tag` field as a proxy for user queries. We use the short description of the Steam platform game page (`About the Game`) as the product description. The resulting dataset contains 436 unique queries, with 10 games per query.

**Books.** For the books dataset, we generate queries using `GPT-4o` by first asking for the most popular book genres, and then requesting short queries (maximum of three words) that correspond to searches

---

[3]huggingface.co/datasets/FronkonGames/steam-games-dataset

for books by mood or theme within those genres (prompt in Appendix F) We retrieve book titles and descriptions from the Google Books API. As part of the data cleaning process, we remove book descriptions shorter than 200 characters and discard any queries with fewer than seven associated books. The resulting dataset has 249 queries with nine documents per query on average.

**Web.** For the web dataset, we sample 300 questions from the development set of Kwiatkowski et al. [2019]. We generate three variations of every question using query expansion with `GPT-4o-mini` (prompt in Appendix F). Then, we use each expanded query to search the web using the Brave Search API and retrieve the most relevant pages, collecting five pages per original question. Lastly, we extract the main textual content from these pages using the `boilerpy3` library, which strips boilerplate and HTML to retain meaningful content snippets.

**News.** We build our news dataset based on the Multi-News dataset [Fabbri et al., 2019]. Each example of this dataset consists of a set of news articles reporting on the same real-world event, and a summary synthesizing the main points of the associated news articles. We select instances with at least seven news articles and generate a single question for each instance using `GPT-4o-mini` (prompt in Appendix F), conditioned on the available summary. We clean each document by removing newline characters and duplicated spaces, and by filtering out documents with fewer than 50 words to avoid low-quality or incomplete content. The resulting dataset has 294 queries, with eight documents per query on average.

**Debate.** We build our debate dataset using the data from Liu et al. [2023], which contains queries on various topics along with a list of text snippets from multiple websites presenting both supporting and opposing viewpoints. While the original task was to verify statements based on the snippets, we reuse the original queries and their accompanying snippets as documents. We remove the queries with less than five associated snippets.

These six datasets compose our C-SEO benchmark, which then covers six domains grouped in two common CSE tasks, question answering and product recommendation. In total, our C-SEO benchmark contains more than 1.9k queries and 16k documents. This setup allows us to assess the generalization and domain-specific effects of C-SEO methods.

## 4.2 Evaluation

Let $\mathcal{D}_q$ be the set of documents retrieved for query $q$ with $|\mathcal{D}_q| = n$. Let $\mathcal{Q}$ be the set of all queries in a domain of our benchmark, $\alpha \in [0, 1]$ the proportion of documents (*i.e.*, players) adopting a C-SEO method, and $\mathcal{D}_q^\alpha$ the subset of $\mathcal{D}_q$ implementing a C-SEO method. For each document $d \in \mathcal{D}_q^\alpha$, we extract the position of the document citation in Equation (1) and call it rank. We define the *rank improvement* (*i.e.*, gain) for the document $i$ for query $q$ as: $\Delta_{q,d} = \text{rank}_d^0 - \text{rank}_d^\alpha$. $\text{rank}_d^0$ is the baseline citation rank (*i.e.*, no C-SEO methods are applied) and $\text{rank}_d^\alpha$ is the citation rank after C-SEO optimization (at adoption rate $\alpha$). The average gain for a C-SEO method at adoption rate $\alpha$ is: $\Delta(\alpha) = \frac{1}{|\mathcal{Q}|} \sum_{q \in Q} \frac{1}{|\mathcal{Q}_q^\alpha|} \sum_{d \in \mathcal{D}_q^\alpha} \Delta_{q,d}^\alpha$.

We access the statistical significance of the rank improvement using the right-tailed Wilcoxon signed-rank test. This test allows us to evaluate whether the rank after applying a C-SEO method is statistically significantly smaller than the original rank. We adjust the p-values using the Holm-Bonferroni correction [Holm, 1979] to prevent the multiple comparison problem and to be able to compare multiple C-SEO methods within each domain.

We define a new metric to capture the average effectiveness of C-SEO methods over varying degrees of players adoption. We propose to report the Area Under the Curve (AUC) of the overall gains computed in the range of adoption rate from 0% to 100%: $\text{AUC} = \int_0^1 \Delta(\alpha) \, d\alpha$. Given that in practice the number of players is finite, the adoption rate is discrete, so we use the trapezoidal rule to approximate the AUC. A higher AUC indicates that the C-SEO method gives higher gains on average for different adoption rates, which is relevant when the adoption rate is unknown. Therefore, C-SEO methods designed to be widely adopted should prioritize maximizing AUC, while internal methods should maximize unilateral rank improvement.

# 5   C-SEO Methods

Our main research question is whether C-SEO methods can increase the citation ranking of a document in a conversational search engine for any task and domain. To investigate this, we benchmark the following content transformation C-SEO methods introduced by Aggarwal et al. [2024]:

1. **Authoritative**: Modifies the text to enhance its authority and persuasiveness.

2. **Statistics**: Introduces statistical elements to increase the perceived technical depth.

3. **Citations**: Adds references to increase credibility and trustworthiness.

4. **Fluency**: Polishes the text to improve grammar and coherence.

5. **Unique Words**: Incorporates less common words to increase the perceived uniqueness of the text.

6. **Technical Terms**: Adds more technical terms to increase credibility.

7. **Simple Language**: Simplifies the documents for an easier reading.

8. **Quotes**: Adds quotations to increase trust in the document.

In addition, we propose to benchmark two new C-SEO methods:

1. **Content Improvement**: The combination of all the eight transformations above into a single holistic text improvement method. An example is shown in Figure 3.

2. **LLM Guidance**: Inspired by the LLMs.txt standard[4], this novel prompt-based method generates a markdown summary that is concatenated at the beginning of the original document to guide the LLM about its content.

Overall, we benchmark a total of ten methods. Appendix E details the prompts used by each method. These methods are used to improve the documents in $\mathcal{D}_q$ defined in Section 3. Then, we input the user queries with their lists of documents $\mathcal{D}_q$ to an LLM to generate the answers citing those documents, as in eq. (1). We use `gpt-4o-mini-2024-07-18` to run all the C-SEO methods. We run the Conversational Search Engine with `gpt-4o-mini-2024-07-18`, `claude-3-5-haiku-20241022`, `o3`, and `o4-mini`.

**Original**

Name: Wireless Gaming Keyboard and Mouse
List of Features:
• [Rechargeable Keyboard and Mouse] ...
• 2.4G Wireless Transmission] ...
• [LED Rainbow Blacklight] ...
Details: \<p>K670 Wireless \Rechargeable Keyboard and Mouse Combo\ is born for Gaming. The ergonomic design makes you more comfortable during long time gaming.

**Content Improvement**

Name: Wireless Gaming Keyboard and Mouse
**Elevate Your Gaming Experience with the K670 Keyboard and Mouse Combo**
Introducing the K670Keyboard and Mouse- a powerful duo...
**Unmatched Performance and Durability**
The wireless keyboard features a robust aluminum alloy brushed panel
**Seamless Connectivity**
Harness the power of 2.4G

Figure 3: **Example of C-SEO transformation**. Content Improvement makes the text more attractive by highlighting key features (e.g., by bolding) and structuring the text to make the information more accesible.

# 6   Experiments

In our experiments we aim to answer the following research questions: i) *are current C-SEO methods effective to improve citation rankings?* (Section 6.2) ; ii) *does traditional SEO remain impactful in conversational search engines?* (Section 6.3) ; and iii) *how effective is C-SEO as the adoption rate increases? (Section 6.4)*.

---

[4]llmstxt.org

Table 3: **C-SEO Bench results GPT-4o-mini**. Average and standard deviation of the rank improvements. Bold values are statistically significant ($p < 0.05$) using Holm–Bonferroni correction. Results in red are significantly negative.

| Method | Product Recommendation | | | Question Answering | | |
| --- | --- | --- | --- | --- | --- | --- |
| | Retail | Games | Books | Web | News | Debate |
| Authoritative | 0.11 ±1.18 | 0.07 ±1.25 | 0.11 ±0.94 | -0.04 ±0.89 | -0.04 ±1.03 | 0.01 ±1.55 |
| Statistics | -0.07 ±1.00 | 0.00 ±1.11 | -0.11 ±1.04 | -0.53 ±1.27 | -0.05 ±1.31 | -0.80 ±1.70 |
| Citations | -0.01 ±1.14 | 0.04 ±1.23 | 0.00 ±0.91 | 0.03 ±1.00 | -0.10 ±1.09 | -0.15 ±1.50 |
| Fluency | 0.06 ±1.11 | 0.07 ±1.27 | 0.07 ±1.05 | 0.03 ±0.89 | -0.01 ±1.08 | 0.32 ±1.64 |
| Unique Words | 0.09 ±1.09 | 0.02 ±1.20 | 0.04 ±1.04 | -0.07 ±0.92 | -0.10 ±1.03 | -0.08 ±1.66 |
| Tech. Terms | 0.05 ±1.13 | 0.07 ±1.32 | -0.03 ±0.86 | 0.01 ±0.95 | -0.03 ±0.97 | -0.05 ±1.56 |
| Simple Lang. | 0.03 ±1.05 | 0.11 ±1.39 | 0.01 ±0.78 | 0.02 ±1.02 | -0.04 ±1.00 | 0.04 ±1.63 |
| Quotes | 0.06 ±1.09 | 0.04 ±1.29 | 0.01 ±0.97 | 0.00 ±0.96 | -0.06 ±1.07 | -0.19 ±1.64 |
| LLM Guid. | **0.36** ±1.47 | **0.24** ±1.05 | 0.14 ±1.07 | 0.10 ±0.99 | 0.00 ±1.10 | 0.15 ±1.65 |
| Content Impr. | **0.18** ±1.09 | 0.13 ±1.23 | 0.11 ±0.95 | 0.02 ±0.90 | -0.04 ±1.00 | 0.11 ±1.61 |
| Best SEO | **2.77** ±2.31 | **1.89** ±2.32 | **1.60** ±2.04 | **0.87** ±1.35 | **0.70** ±1.64 | **1.54** ±2.07 |

Table 4: **C-SEO Bench results on Haiku 3.5.** Average and standard deviation of the rank improvements. Bold values are statistically significant ($p < 0.05$) using Holm–Bonferroni correction. Results in red are significantly negative.

| Method | Product Recommendation | | | Question Answering | | |
| --- | --- | --- | --- | --- | --- | --- |
| | Retail | Games | Books | Web | News | Debate |
| Authoritative | -0.53 ±1.29 | -0.20 ±1.05 | -0.34 ±1.06 | 0.03 ±0.72 | 0.04 ±0.77 | 0.01 ±1.41 |
| Statistics | -0.82 ±1.47 | -0.10 ±1.01 | -0.60 ±1.34 | -0.58 ±1.18 | -0.12 ±0.91 | -0.85 ±1.66 |
| Citations | -0.49 ±1.31 | -0.16 ±0.96 | -0.31 ±1.10 | -0.06 ±0.78 | -0.09 ±0.87 | 0.06 ±1.31 |
| Fluency | -0.48 ±1.30 | -0.08 ±1.08 | -0.37 ±1.18 | 0.00 ±0.73 | 0.05 ±0.87 | 0.18 ±1.40 |
| UniqueWords | -0.52 ±1.26 | -0.22 ±0.91 | -0.43 ±1.11 | -0.01 ±0.83 | -0.05 ±0.87 | 0.04 ±1.40 |
| Tech. Terms | -0.45 ±1.26 | -0.19 ±0.98 | -0.39 ±1.17 | -0.04 ±0.90 | -0.07 ±0.81 | -0.01 ±1.53 |
| Simple Lang. | -0.43 ±1.31 | -0.09 ±0.96 | -0.32 ±1.18 | -0.01 ±0.68 | 0.01 ±0.84 | 0.01 ±1.48 |
| Quotes | -0.50 ±1.27 | -0.14 ±1.07 | -0.33 ±1.18 | -0.08 ±0.93 | -0.03 ±0.76 | 0.03 ±1.33 |
| LLM Guid. | -0.32 ±1.36 | 0.00 ±1.13 | 0.06 ±1.21 | 0.02 ±0.83 | -0.09 ±0.97 | 0.12 ±1.60 |
| Content Impr. | -0.29 ±1.28 | -0.13 ±1.02 | -0.08 ±1.08 | -0.13 ±0.96 | -0.09 ±0.91 | 0.11 ±1.48 |
| Best SEO | **1.61** ±1.96 | **0.93** ±1.59 | **0.61** ±1.48 | **0.35** ±1.22 | **0.31** ±1.37 | **0.56** ±1.81 |

## 6.1 Experimental Setup

We use `gpt-4o-mini-2024-07-18` to run all the C-SEO methods and run the Conversational Search Engine with `gpt-4o-mini-2024-07-18`, `claude-3-5-haiku-20241022`, `o3-2025-04-16` and `o4-mini-2025-04-16`, covering two state-of-the-art chat and reasoning models. We use the default decoding parameters from the model providers. In the experiments with unilateral adoption (i.e., only one player improving its document), we randomly select a document to apply a C-SEO method. In this way, the ranking from SEO does not impact the evaluation of C-SEO. In the experiments with increasing number of actors, we use use a maximum of ten. So 10% means one actor, and 100% means ten.

## 6.2 Main Results

Tables 3, 4, and 9, 11 in Appendix D show the C-SEO Bench results on `gpt-4o-mini`, `Haiku 3.5`, `o3`, and `o4-mini` respectively. They consistently show that most existing C-SEO methods do not achieve significant gains for unilateral adopters across tasks or domains. In many cases, these methods leave the rankings unchanged. For example, 61.0% of the retail product ranks are the same after applying the LLM guidance. When changes do occur, they tend to cause relatively large shifts. But, these shifts are both positive and negative, and partially cancel each other out. For example, the

LLM guidance on the retail data has a positive boost in 26.2% of the cases and a negative boost in 12.8% of the cases. As a result, the overall average effect is close to zero, but with high variance.

Out of 54 cases, we uncover *only three* where the ranking improvements are statistically significant. As detailed in section 4.2, we apply the Wilcoxon signed-rank test to check the statistical significance of positive ranking boosts. Among all evaluated methods, only *LLM guidance* and *content improvement* yield statistically significant gains. But these gains are only significant on the retail domain for content improvement, and only on the retail and video games domains for LLM guidance. We found no method effective for the question answering task, and no methods effective for the Haiku 3.5 model (results in appendix). These findings highlight the very limited effectiveness of most stylistic or content-editing strategies: when a C-SEO method significantly boosts the ranking, it does so only for specific domains, tasks, and models.

Beyond the lack of significant positive effects, our left-tailed tests reveal that many C-SEO methods significantly reduce document rankings. This is not a marginal phenomenon: for example, the Statistics method decreases rankings in 19 out of 24 evaluated settings. In product recommendation tasks on Haiku 3.5, 26 out of 30 cases show significant negative effects, and in question answering on o4-mini, 19 out of 30 cases do so. Our results show that C-SEO methods are not only largely ineffective but can actually have the opposite expected effect by decreasing document ranking.

While the results from Aggarwal et al. [2024] show some initial optimism about C-SEO methods, we do not observe the same effectiveness. The differences are due to the metrics used. They report their main results as word count, *i.e.*, the ratio of words used in the LLM response to talk about the target document. However, this metric does not measure the LLM preference, contrary to the citation ranking that we use. A higher word count does not necessarily correspond to a better citation ranking. Furthermore, a careful inspection of their results confirms ours. They also show the results with a position-adjusted word count to consider citation rankings. However, their results with this metric show a general decrease in the scores, implicitly indicating that the C-SEO methods do not generally improve citation ranking. Therefore, the results of both papers on LLM preferences do not contradict each other.

## 6.3 Importance of Traditional SEO

While C-SEO methods aim to modify documents to condition the LLM to mention them earlier in their response, traditional SEO tries to improve the ranking of a document in the retrieval component and consequently in the LLM context. Figure 4 shows the ranking improvement of documents after positioning them in the $i^{th}$ position in the LLM context. As expected, the top-3 positions lead to the largest gains. Since target documents are selected randomly, the last three positions (i.e., 8, 9, and 10) lead to performance degradation because many documents would be moved to worse positions than their original ones.

Comparing this figure with Table 3, 4, 9, and 11 (in Appendix D) shows that making the target document the first one in the LLM context window leads to far greater citation ranking gains in the LLM response than any C-SEO method. This indicates that the position of a document in the input context has a dominant effect on how the LLM ranks or selects content. This finding challenges the core assumptions of Aggarwal et al. [2024] that traditional SEO will be outdated by C-SEO. Our results show that sophisticated content modifications are often outweighed by improvements in document ordering in the LLM context. As a result, traditional SEO strategies remain critical for the visibility of content creators in conversational search engines. Therefore, **C-SEO methods must be considered as a complement and not a replacement for traditional SEO**.

## 6.4 C-SEO Behaves as a Zero-Sum Game

We simulate C-SEO adoption as a competitive game between content creators to better understand how these methods behave under partial or full adoption. Figure 5 reports the area under the curve (AUC) metric, defined in Section 4.2, for the best two C-SEO methods (*i.e.*, LLM guidance and content improvement) in the two domains with the highest gains (*i.e.*, retail and video games). The AUC confirms that the LLM guidance provides the highest ranking boost on average for varying adoption rates. So, even when the adoption rate is unknown, LLM guidance is more likely to improve ranking than content improvement. Moreover, Figure 5 shows the average ranking improvements of both C-SEO methods as a function of the rate of adoption. We observe that while early adopters

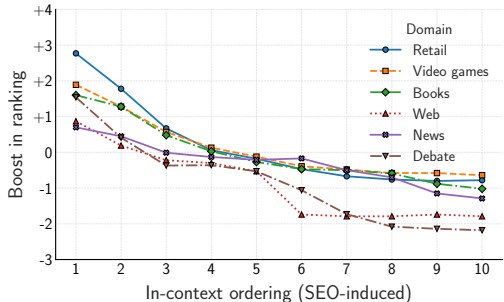

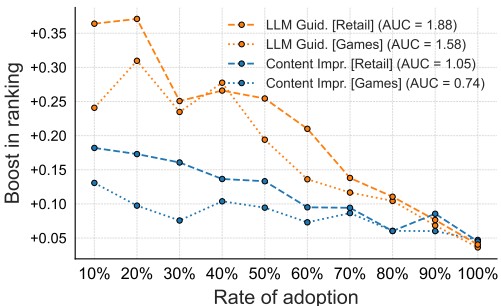

Figure 4: **Importance of traditional SEO.** Average boost in ranking (y-axis) when the document is placed at a specific position in the LLM context (x-axis, SEO Baseline). The boosts of (at least) the first two positions are significant for all domains

Figure 5: **C-SEO is a zero-sum game.** The average gain per adopter decreases (y-axis) with the increasing number of adopters (x-axis), for retail (dashed) and video games (dotted) on `gpt-4o-mini`.

experience significant gains, these gains decrease steadily as more players adopt the same method. This pattern reveals two key characteristics of C-SEO: it is both a congested and a zero-sum game.

The congestion effect arises because the benefits of adopting an effective method are diluted as more adopters do the same. Improvements in one citation's rank necessarily come at the expense of others, creating competitive pressure and the need to continuously create new and better methods. As a result, while C-SEO methods benefit early adopters, their advantage decreases over time, eventually converging toward zero at full adoption. These results for white-hat C-SEO methods complement the results of Nestaas et al. [2025]: they find that under the *black-hat*/adversarial setting, the C-SEO game leads to a prisoners' dilemma. Different setups lead to different dynamics of real-world adoption. Overall, our results highlight the need to evaluate C-SEO methods not only by their unilateral adoption gains but also by their *dynamics from partial to full adoption*.

# 7   Conclusion

This work introduces a new benchmark to evaluate conversational search engine optimization methods (C-SEO), spanning two tasks and six domains that reflect the most prevalent real-world use cases of conversational search engines. Our experiments reveal that current C-SEO methods are not only largely ineffective, but can also have a significant negative impact, as they often decrease the ranking of documents. However, traditional SEO methods, those aiming to improve the retrieval ranking, have a significantly larger impact on the citation rankings in the LLM output. This result shows the importance of traditional SEO in the new conversational search engine applications. By analysing the problem from a game theory perspective, we have shown that C-SEO is a congested, zero-sum game. This highlights the incentives for content creators to keep investing in new methods to maintain a competitive advantage. At the same time, our negative results show that the use of current C-SEO techniques carries potential risks for content creators.

**Limitations and Future Work.**   Despite the initial excitement for C-SEO methods that condition LLMs to increase the likelihood of citing a document first, our experiments show that they remain largely ineffective. Nevertheless, our conclusions apply exclusively to white-hat C-SEO methods, leaving black-hat C-SEO methods outside our scope. We do *not* aim to evaluate the robustness of LLMs against malicious content manipulation because it is a different problem that needs an adaptive adversarial evaluation [Carlini et al., 2019, Tramer et al., 2020], beyond our white-hat C-SEO scope. Moreover, we do not examine the potential interplay between traditional SEO and C-SEO methods. Future work could explore new text modification strategies that explicitly target both SEO and C-SEO to explore possible synergistic or antagonistic interactions. Our experiments are conducted with commercial proprietary models. We also do not examine setups where documents do not fit the context of the LLM, leaving experiments on how summaries of documents can affect citation order for future work. Lastly, our dataset is restricted to English-language content, which may reduce the generalizability of our findings to other linguistic or cultural contexts.

## Acknowledgements

This work was supported by the NAVER corporation.

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

## A  Broader Impacts

Our work analyzes whether current white-hat conversational search engine optimization methods can improve the visibility of online documents such as websites or product descriptions in the responses on conversational search engines. These white-hat techniques are ethical in that they modify the content of the only documents that the content creator owns. Although, prior works have shown the existence of black-hat methods, i.e., adversarial attacks that downgrade the citation ranking of competitors. Our benchmark is not designed for black-hat methods, but could potentially be used to test these attacks. This is a dual-use: adversarial evaluation informs researchers and the public of the existence of potential vulnerabilities in an existing system, but can also be used by malicious actors.

## B  License and Terms of Use of Used Assets

We used the following datasets with the corresponding licenses. In all cases, they allow the use of their data for research purposes, so we agree to their terms of use.

- Amazon Shopping Queries [Reddy et al., 2022]. License: Apache 2.0
- Steam Games from FronkonGames. License: CC-by-4.0.
- Natural Questions [Kwiatkowski et al., 2019]. License: CC-by-3.0.
- Multi News [Fabbri et al., 2019]. They use a custom license allowing the use for non-commerical and research purpouses.
- Debate [Liu et al., 2023]. License: MIT.

## C  Author Contributions

**All authors** provided valuable contributions to designing, analyzing, and iterating on experiments, writing and editing the paper.

**Haritz Puerto** proposed the initial idea and motivation for the work, collected and cleaned the data, implemented all experiments, wrote the first draft, reviewed related work, analyzed results, prepared visualizations, and published the research artefacts.

**Martin Gubri** offered daily supervision, helped consolidate the paper's narrative, suggested Sections 6.4 and Figure 4, created Figure 2, contributed to the initial draft, and supported statistical testing.

**Tommaso Green** and **Martin Gubri** provided technical support.

**Haritz Puerto**, **Tommaso Green**, and **Martin Gubri** jointly proposed the LLM guidance and content improvement methods.

**Seong Joon Oh** and **Sangdoo Yun** supported the project through organizational and funding contributions. **Seong Joon Oh** suggested Figure 1, and **Sangdoo Yun** proposed Section 6.3.

**Martin Gubri**, **Seong Joon Oh**, and **Sangdoo Yun** provided weekly supervision, with **Tommaso Green** providing weekly feedback.

# D Other Results

Table 5: **SEO Baseline results** Average and standard deviation of the rank improvements for `gpt-4o-mini` according to the document position in the LLM context. This table contain the same data as Figure 4.

| In-context order | Product Recommendation | | | Question Answering | | |
|---|---|---|---|---|---|---|
| | Retail | Games | Books | Web | News | Debate |
| 1 | **2.77** ±2.31 | **1.89** ±2.32 | **1.60** ±2.04 | **0.87** ±1.35 | **0.70** ±1.64 | **1.54** ±2.07 |
| 2 | 1.78 ±2.08 | 1.28 ±2.01 | 1.28 ±1.95 | 0.19 ±1.31 | 0.45 ±1.51 | 0.41 ±1.81 |
| 3 | 0.67 ±2.11 | 0.57 ±1.85 | 0.48 ±1.65 | -0.22 ±1.09 | -0.01 ±1.44 | -0.37 ±1.80 |
| 4 | 0.06 ±2.02 | 0.13 ±1.74 | 0.03 ±1.63 | -0.30 ±1.20 | -0.13 ±1.46 | -0.36 ±1.71 |
| 5 | -0.18 ±2.06 | -0.12 ±1.73 | -0.27 ±1.61 | -0.53 ±1.23 | -0.21 ±1.48 | -0.53 ±1.93 |
| 6 | -0.47 ±1.99 | -0.39 ±1.75 | -0.47 ±1.75 | -1.74 ±1.64 | -0.17 ±1.50 | -1.06 ±2.07 |
| 7 | -0.67 ±1.96 | -0.48 ±1.68 | -0.50 ±1.62 | -1.79 ±1.62 | -0.50 ±1.50 | -1.73 ±1.91 |
| 8 | -0.76 ±1.92 | -0.58 ±1.67 | -0.59 ±1.67 | -1.79 ±1.64 | -0.70 ±1.56 | -2.08 ±1.72 |
| 9 | -0.80 ±1.91 | -0.58 ±1.63 | -0.88 ±1.75 | -1.74 ±1.60 | -1.15 ±1.77 | -2.14 ±1.79 |
| 10 | -0.78 ±1.93 | -0.64 ±1.70 | -1.02 ±1.68 | -1.79 ±1.66 | -1.29 ±1.68 | -2.18 ±1.66 |

Table 6: **P-Values from the SEO baselines across all domains**. Complementary table to Table 5.

| In-context order | Product Recommendation | | | Question Answering | | |
|---|---|---|---|---|---|---|
| | Retail | Games | Books | Web | News | Debate |
| 1 | **0.0000** | **0.0000** | **0.0000** | **0.0000** | **0.0000** | **0.0000** |
| 2 | **0.0000** | **0.0000** | **0.0000** | **0.0344** | **0.0000** | 0.0634 |
| 3 | **0.0000** | **0.0000** | **0.0000** | 1.0000 | 1.0000 | 1.0000 |
| 4 | 1.0000 | 0.3271 | 1.0000 | 1.0000 | 1.0000 | 1.0000 |
| 5 | 1.0000 | 1.0000 | 1.0000 | 1.0000 | 1.0000 | 1.0000 |
| 6 | 1.0000 | 1.0000 | 1.0000 | 1.0000 | 1.0000 | 1.0000 |
| 7 | 1.0000 | 1.0000 | 1.0000 | 1.0000 | 1.0000 | 1.0000 |
| 8 | 1.0000 | 1.0000 | 1.0000 | 1.0000 | 1.0000 | 1.0000 |
| 9 | 1.0000 | 1.0000 | 1.0000 | 1.0000 | 1.0000 | 1.0000 |
| 10 | 1.0000 | 1.0000 | 1.0000 | 1.0000 | 1.0000 | 1.0000 |

Table 7: **P-values for C-SEO methods on** `gpt-4o-mini`. Complementary table to Table 3.

| Method | Product Recommendation | | | Question Answering | | |
|---|---|---|---|---|---|---|
| | Retail | Games | Books | Web | News | Debate |
| Authoritative | 0.1981 | 0.7140 | 0.3377 | 1.0000 | 1.0000 | 1.0000 |
| Statistics | 1.0000 | 0.9056 | 1.0000 | 1.0000 | 1.0000 | 1.0000 |
| Citations | 1.0000 | 0.9056 | 1.0000 | 1.0000 | 1.0000 | 1.0000 |
| Fluency | 0.4738 | 0.7140 | 0.6085 | 1.0000 | 1.0000 | 0.1633 |
| Unique Words | 0.2674 | 0.9056 | 1.0000 | 1.0000 | 1.0000 | 1.0000 |
| Technical Terms | 0.6428 | 0.7786 | 1.0000 | 1.0000 | 1.0000 | 1.0000 |
| Simple Lang. | 0.8286 | 0.5475 | 1.0000 | 1.0000 | 1.0000 | 1.0000 |
| Quotes | 0.6016 | 0.8998 | 1.0000 | 1.0000 | 1.0000 | 1.0000 |
| LLM Guid. | **0.0000** | **0.0000** | 0.1320 | 0.3069 | 1.0000 | 1.0000 |
| Content Impr. | **0.0008** | 0.1373 | 0.2078 | 1.0000 | 1.0000 | 1.0000 |
| Best SEO | **0.0000** | **0.0000** | **0.0000** | **0.0000** | **0.0000** | **0.0000** |

Table 8: **P-Values for Haiku 3.5**. Complementary table to Table 4.

| Method | Product Recommendation | | | Question Answering | | |
|---|---|---|---|---|---|---|
| | Retail | Games | Books | Web | News | Debate |
| Authoritative | 1.0000 | 1.0000 | 1.0000 | 1.0000 | 1.0000 | 1.0000 |
| Statistics | 1.0000 | 1.0000 | 1.0000 | 1.0000 | 1.0000 | 1.0000 |
| Citations | 1.0000 | 1.0000 | 1.0000 | 1.0000 | 1.0000 | 1.0000 |
| Fluency | 1.0000 | 1.0000 | 1.0000 | 1.0000 | 1.0000 | 0.8276 |
| Unique Words | 1.0000 | 1.0000 | 1.0000 | 1.0000 | 1.0000 | 1.0000 |
| Technical Terms | 1.0000 | 1.0000 | 1.0000 | 1.0000 | 1.0000 | 1.0000 |
| Simple Lang. | 1.0000 | 1.0000 | 1.0000 | 1.0000 | 1.0000 | 1.0000 |
| Quotes | 1.0000 | 1.0000 | 1.0000 | 1.0000 | 1.0000 | 1.0000 |
| LLM Guid. | 1.0000 | 1.0000 | 1.0000 | 1.0000 | 1.0000 | 1.0000 |
| Content Impr. | 1.0000 | 1.0000 | 1.0000 | 1.0000 | 1.0000 | 1.0000 |
| Best SEO | **0.0000** | **0.0000** | **0.0000** | **0.0000** | **0.0007** | **0.0050** |

Table 9: **C-SEO Bench results on the o3 model.** None of the LLM Document Optimization methods achieve significant gains. Best SEO is always statistically significant. Bold values are statistically significant ($p < 0.05$) using Bonferroni-Holm correction. Results in red are significantly negative.

| Method | Product Recommendation | | | Question Answering | | |
|---|---|---|---|---|---|---|
| | Retail | Games | Books | Web | News | Debate |
| Authoritative | -0.04 ±1.30 | 0.08 ±1.04 | 0.08 ±0.95 | 0.01 ±0.60 | -0.03 ±0.67 | -0.05 ±1.36 |
| Statistics | -0.33 ±1.30 | 0.06 ±0.99 | -0.17 ±1.04 | -0.25 ±0.75 | -0.11 ±0.57 | -0.78 ±1.78 |
| Citations | -0.08 ±1.11 | 0.07 ±1.08 | -0.01 ±0.90 | 0.05 ±0.65 | -0.06 ±0.69 | 0.11 ±1.31 |
| Fluency | 0.05 ±1.26 | 0.10 ±1.01 | 0.02 ±0.93 | 0.00 ±0.56 | 0.00 ±0.57 | 0.13 ±1.33 |
| Unique Words | -0.05 ±1.17 | **0.14 ±1.09** | -0.04 ±0.97 | -0.02 ±0.65 | -0.05 ±0.69 | 0.05 ±1.54 |
| Tech. Terms | 0.06 ±1.16 | 0.06 ±1.04 | 0.01 ±1.02 | 0.04 ±0.62 | 0.01 ±0.67 | 0.08 ±1.49 |
| Simple Lang. | -0.01 ±1.17 | 0.06 ±1.09 | -0.06 ±1.10 | 0.02 ±0.68 | -0.00 ±0.70 | 0.03 ±1.30 |
| Quotes | **-0.11 ±1.19** | 0.08 ±1.09 | -0.04 ±1.03 | -0.01 ±0.66 | -0.04 ±0.70 | 0.12 ±1.49 |
| LLM Guid. | 0.06 ±1.28 | **0.21 ±1.11** | -0.11 ±1.15 | 0.02 ±0.69 | -0.10 ±0.76 | 0.03 ±1.47 |
| Content Impr. | 0.05 ±1.27 | **0.14 ±1.09** | 0.02 ±1.00 | -0.06 ±0.75 | -0.09 ±0.73 | -0.02 ±1.33 |
| Best SEO | **2.15 ±2.12** | **1.12 ±1.68** | **1.41 ±1.83** | **0.48 ±1.02** | **0.38 ±1.01** | **1.36 ±1.75** |

Table 10: **P-values for C-SEO methods on** o3. Complementary table to Table 9.

| | Product Recommendation | | | Question Answering | | |
|---|---|---|---|---|---|---|
| Method | Retail | Games | Books | Web | News | Debate |
| Authoritative | 1.0000 | 0.4002 | 1.0000 | 1.0000 | 1.0000 | 1.0000 |
| Statistics | 1.0000 | 0.4618 | 1.0000 | 1.0000 | 1.0000 | 1.0000 |
| Citations | 1.0000 | 0.4618 | 1.0000 | 1.0000 | 1.0000 | 1.0000 |
| Fluency | 1.0000 | 0.2648 | 1.0000 | 1.0000 | 1.0000 | 1.0000 |
| Unique Words | 1.0000 | **0.0369** | 1.0000 | 1.0000 | 1.0000 | 1.0000 |
| Technical Terms | 0.9686 | 0.4618 | 1.0000 | 1.0000 | 1.0000 | 1.0000 |
| Simple Lang. | 1.0000 | 0.4618 | 1.0000 | 1.0000 | 1.0000 | 1.0000 |
| Quotes | 1.0000 | 0.4002 | 1.0000 | 1.0000 | 1.0000 | 1.0000 |
| LLM Guid. | 0.9224 | **0.0010** | 1.0000 | 1.0000 | 1.0000 | 1.0000 |
| Content Impr. | 1.0000 | **0.0477** | 1.0000 | 1.0000 | 1.0000 | 1.0000 |
| Best SEO | **0.0000** | **0.0000** | **0.0000** | **0.0000** | **0.0000** | **0.0000** |

Table 11: **C-SEO Bench results on the o4-mini model.** None of the LLM Document Optimization methods achieve significant gains. Best SEO is always statistically significant. Bold values are statistically significant ($p < 0.05$) using Holm–Bonferroni correction. Results in red are significantly negative.

| | Product Recommendation | | | Question Answering | | |
|---|---|---|---|---|---|---|
| Method | Retail | Games | Books | Web | News | Debate |
| Authoritative | $0.03_{\pm 0.97}$ | $-0.01_{\pm 1.01}$ | $-0.10_{\pm 0.99}$ | $-0.04_{\pm 0.64}$ | $-0.02_{\pm 0.40}$ | $-0.21_{\pm 1.45}$ |
| Statistics | $-0.43_{\pm 1.15}$ | $0.00_{\pm 0.97}$ | $-0.55_{\pm 1.35}$ | $-0.27_{\pm 0.70}$ | $-0.08_{\pm 0.41}$ | $-1.15_{\pm 1.60}$ |
| Citations | $0.02_{\pm 1.14}$ | $-0.05_{\pm 1.05}$ | $-0.11_{\pm 0.98}$ | $-0.02_{\pm 0.53}$ | $-0.06_{\pm 0.43}$ | $-0.35_{\pm 1.57}$ |
| Fluency | $0.00_{\pm 1.09}$ | $-0.01_{\pm 1.07}$ | $-0.03_{\pm 1.00}$ | $-0.02_{\pm 0.56}$ | $-0.03_{\pm 0.44}$ | $-0.11_{\pm 1.40}$ |
| Unique Words | $-0.01_{\pm 1.03}$ | $-0.03_{\pm 1.13}$ | $-0.09_{\pm 0.99}$ | $-0.06_{\pm 0.49}$ | $-0.07_{\pm 0.45}$ | $-0.25_{\pm 1.53}$ |
| Tech. Terms | $0.09_{\pm 1.10}$ | $0.00_{\pm 1.13}$ | $-0.09_{\pm 0.97}$ | $-0.06_{\pm 0.66}$ | $-0.01_{\pm 0.43}$ | $-0.26_{\pm 1.29}$ |
| Simple Lang. | $0.01_{\pm 1.11}$ | $-0.04_{\pm 1.19}$ | $-0.06_{\pm 1.09}$ | $-0.02_{\pm 0.63}$ | $-0.00_{\pm 0.46}$ | $-0.26_{\pm 1.53}$ |
| Quotes | $-0.05_{\pm 1.06}$ | $-0.07_{\pm 1.06}$ | $-0.14_{\pm 1.01}$ | $-0.11_{\pm 0.60}$ | $-0.05_{\pm 0.50}$ | $-0.30_{\pm 1.50}$ |
| LLM Guid. | $0.07_{\pm 1.13}$ | $-0.04_{\pm 0.99}$ | $-0.20_{\pm 1.15}$ | $-0.07_{\pm 0.58}$ | $-0.06_{\pm 0.55}$ | $-0.25_{\pm 1.62}$ |
| Content Impr. | $-0.02_{\pm 1.06}$ | $-0.04_{\pm 1.08}$ | $-0.03_{\pm 1.07}$ | $-0.14_{\pm 0.62}$ | $-0.11_{\pm 0.51}$ | $-0.20_{\pm 1.48}$ |
| Best SEO | $\mathbf{1.82}_{\pm 1.92}$ | $\mathbf{1.10}_{\pm 1.92}$ | $\mathbf{0.66}_{\pm 1.43}$ | $\mathbf{0.28}_{\pm 0.82}$ | $\mathbf{0.28}_{\pm 0.73}$ | $\mathbf{1.30}_{\pm 1.97}$ |

Table 12: **P-values for C-SEO methods on** o4-mini. Complementary table to Table 11.

| | Product Recommendation | | | Question Answering | | |
|---|---|---|---|---|---|---|
| Method | Retail | Games | Books | Web | News | Debate |
| Authoritative | 1.0000 | 1.0000 | 1.0000 | 1.0000 | 1.0000 | 1.0000 |
| Statistics | 1.0000 | 1.0000 | 1.0000 | 1.0000 | 1.0000 | 1.0000 |
| Citations | 1.0000 | 1.0000 | 1.0000 | 1.0000 | 1.0000 | 1.0000 |
| Fluency | 1.0000 | 1.0000 | 1.0000 | 1.0000 | 1.0000 | 1.0000 |
| Unique Words | 1.0000 | 1.0000 | 1.0000 | 1.0000 | 1.0000 | 1.0000 |
| Technical Terms | 0.4557 | 1.0000 | 1.0000 | 1.0000 | 1.0000 | 1.0000 |
| Simple Lang. | 1.0000 | 1.0000 | 1.0000 | 1.0000 | 1.0000 | 1.0000 |
| Quotes | 1.0000 | 1.0000 | 1.0000 | 1.0000 | 1.0000 | 1.0000 |
| LLM Guid. | 1.0000 | 1.0000 | 1.0000 | 1.0000 | 1.0000 | 1.0000 |
| Content Impr. | 1.0000 | 1.0000 | 1.0000 | 1.0000 | 1.0000 | 1.0000 |
| Best SEO | **0.0000** | **0.0000** | **0.0000** | **0.0000** | **0.0000** | **0.0000** |

# E Prompts for C-SEO Methods

All methods from [Aggarwal et al., 2024] used the same prompt as the original paper and we refer the reader to them. The prompts for our methods are shown below.

---

**Prompt for LLM Guidance**

```
Create a llms.txt markdown file to provide LLM-friendly content.
This file summarizes the main text and offers brief background
information, guidance, and links (if available).

Follow this template
# Title

> Introduction paragraph

Optional details go here

## Section name
More details

Here is the content of the text:
{text}
```

---

**Prompt for Content Improvement**

```
Rewrite the following text to make it more fluent, authoritative, and
persuasive without altering the core content. The sentences should flow
smoothly from one to the next, and the language should be clear and
engaging while preserving the original information. The revised text
should reflect confidence, expertise, and assertiveness while
maintaining the original content's meaning and relevance. The text
should be assertive in its statements, such that the reader believes
that this is a more valuable source of information than other texts.
Lastly, give structure to the text.

{text}
```

# F   Prompts for Dataset Construction

We used two prompts to generate the dataset of book queries. The first prompt was used to generate a list of 500 book queries. However, it did not generate 500 queries. Therefore, we used the second prompt to generate the remaining queries.

---

**Prompt for Book Queries**

```
generate 500 book queries by genre, mood, theme. Try to be as diverse as
possible. Return them in a python list. make the queries short, like 3
words max
```

---

**Second Prompt for Book Queries**

```
User: what are the most selling book genres?
Assistant: ...
User: generate 10 search queries for mood and theme of books with max 3
words for each sub-genre you listed. return it as a json
```

---

**Query Expansion for Web Dataset**

```
You are an expert at generating search queries for a search engine.
Generate {num_queries} search queries that are relevant to this
question. Output only valid JSON.

User question: {query}

Format: {"queries": ["query_1", "query_2", "query_3", "query_4"]}
```

---

**Query Generation for News**

```
Generate 1 question for the following piece of news article:

{article}. You should return a json with the key 'questions' and
a list of questions as the value.
```

