# OpenReview forum: "C-SEO Bench: Does Conversational SEO Work?"
_NeurIPS.cc/2025/Datasets_and_Benchmarks_Track — NeurIPS 2025 Datasets and Benchmarks Track poster_

### Official Review · Reviewer_wqWW · 2025-07-02

**Rating:** 5
**Confidence:** 3

**Summary:**

This paper introduces a comprehensive benchmark to evaluate Conversational Search Engine Optimization (C-SEO) across multiple tasks, domains, and number of adopting actors. It is motivated by the growing use of LLMs in search engines, which changes how content is surfaced and cited. The authors evaluate a variety of white-hat C-SEO strategies and compare the effectivness with traditional SEO strategies. The authors find that C-SEO is challenging and the current C-SEO methods are largely ineffective.

**Dataset Code Accessibility:**

Yes

**Ethical Considerations:**

No, there are no or only very minor ethics concerns

**Final Justification:**

I am not familiar with SEO, but the authors have addressed my concerns in the rebuttal, which has increased my confidence in my rating.

**Limitations Weaknesses:**

Most C-SEO methods are significantly weaker than the best SEO, and the results reported in Table 3 may lack robustness. While the findings and insights of this paper are valuable, it remains unclear whether the proposed benchmark is truly well-suited for evaluating C-SEO methods.

**Strengths Contributions:**

1. The C-SEO benchmark is well motivated and meaningful in the LLM era. The proposed C-SEO Bench is comprehensive.
2. The open-source data and code help with reproducibility and community benchmarking.
3. The findings in this paper are insightful. It has been shown that C-SEO methods have limited or domain-specific effect and traditional SEO (document position in context) greatly dominates in influencing citation ranking. It also highlights that the zero-sum nature of C-SEO where increasing adoption reduces individual gains. These insights are crucial for future research and C-SEO strategy design.

---

> ### Author Rebuttal · Authors · 2025-07-29
>
> Thank you for your review. We are glad you find our benchmark well motivated, meaningful and comprehensive, our findings insightful and crucial for future research, and our open-source data and code useful.
>
>
> Answers to the weaknesses
>
> * “Most C-SEO methods are significantly weaker than the best SEO”
>
> This is exactly the key finding we highlight. Our main contribution is to challenge the belief in the community that “C-SEO methods work and will replace traditional SEO.” As we show in the paper, most C-SEO methods do not work, while traditional SEO remains effective in conversational search engines.
>
> * “Table 3 may lack robustness”
>
> Table 3 provides the mean and standard deviation of the performance gains of the C-SEO methods and the best SEO method. Statistically significant results (p < 0.05, Bonferroni-Holm corrected) are bolded, and all p-values are included in Table 8 for transparency. The results show that most C-SEO methods offer no consistent, significant improvements, whereas SEO does for all datasets in our benchmark. These conclusions are consistent with results from other models, including Claude Haiku 3.5 (Table 4) and the reasoning models OpenAI o3 and o4-mini (answer to reviewer 73q6).
>
> * “It remains unclear whether the proposed benchmark is truly well-suited for evaluating C-SEO methods.”
>
> Our benchmark is the first comprehensive and realistic benchmark for C-SEO methods. Prior evaluations were performed on limited setups such as single applications in specific domains and with a single actor adopting the method (lines 42-43). Our benchmark covers more realistic elements in C-SEO application scenarios with 6 domains (retail, news, books, games, web questions, and debates, §4) and varying numbers of adopters (lines 46-60). The conclusion of our work is indeed rather negative: existing C-SEO methods do not work.  However, we believe this will provide a sober realisation for the community that we should rethink the way we build C-SEO methods. We provide a solid groundwork for the community to start the journey to truly effective C-SEO.

---

> > ### Author Response · Authors · 2025-08-05
> >
> > Dear Reviewer wqWW,
> >
> > Thank you for your time reviewing our paper. We hope our rebuttal addressed your concerns. Please let us know if you have any further questions.
> >
> > Thank you.

---

> > > ### Comment · Reviewer_wqWW · 2025-08-08
> > >
> > > Dear authors,
> > >
> > > Thank you for addressing my concerns. I have raised my confidence score correspondingly.

---

### Official Review · Reviewer_73q6 · 2025-07-04

**Rating:** 4
**Confidence:** 5

**Summary:**

The authors propose a dataset consisting of 16k query-documents pairs for domains: Retail, Video Games, Books, Web, News, Debate. The dataset is constructed to study ways of optimization for Conversational Search Engines (C-SEO). Two novel methods are proposed and are compared to 7 existing ones. The work contains description of eval methodology and experiments. The authors also study dependence of C-SEO adoption among participants (i.e. documents). All C-SEO modifications

**Additional Feedback:**

- Lines 125-126 Eq (1) is not clearly defined. What is t_i? Is it a word, or a token?


- Line 126: Should “C_i” be replaced by “c_i” <- small char


- Table 1: there is 10th method called: “Tech. Terms”, this method is not described in the text (there is only 9 methods)

**Dataset Code Accessibility:**

Yes

**Ethical Considerations:**

No, there are no or only very minor ethics concerns

**Limitations Weaknesses:**

-	Not breakthrough results, more incremental (yes, the dataset size is 10 times larger than from previous works..  Is it a great gamechanger? I would think of 100 times or so)

-	I doubt in conclusions in Main Results. The authors claims that 3 experimental configurations (method,dataset) demonstrate statistical significance of improvement, and, thus their novel methods “demonstrate promising results”…. BUT let consider the definition of p-value. P-value is the probability of obtaining test results at least as extreme as the result actually observed, under the assumption that the null hypothesis is correct. In other words, in the case when gain expectation is zero, it is EXPECTED to observe 5% of p-values under <0.05 (because p-value is uniformly distributed over [0,1] under the null hypothesis). So, in the table, we observe 10 * 6 configurations (method,dataset) => 60 calculations of p-value. It is OK to have 3  (5% out of 60) where p-value < 0.05 when gain expectation is zero. So, I would recommend to remove the claim of superiority (“promising”) of the novel method, or change experiment methodology to give more evidence from statistics..

**Strengths Contributions:**

Important area of research

Potentially very applicable

Clear motivation

---

> ### Author Rebuttal · Authors · 2025-07-29
>
> Thank you for your review. We are excited you found our paper covering an important area of research, very applicable, and with a clear motivation.
>
> Answers to weaknesses.
>
> 1. “No breakthrough results” in relation to dataset size
>
> The key breakthrough does not lie in the dataset scale but in challenging a community assumption: “C-SEO methods work and will substitute traditional SEO.” We did not need a 100x larger dataset to make this case; a 10x increase with sufficient diversity was enough to show that current C-SEO methods do not work, and instead, traditional SEO remains effective. We will make this clearer in the paper.
>
>
>
> 2. “Statistical significance of improvement of methods”
>
> This point was considered during our experimental design and analysis. We anticipated the multiple comparisons issue and addressed it by applying the Bonferroni-Holm correction [1] (lines 221–223, Table 3 caption) to control the family-wise error rate at 5% by decreasing the individual significant threshold. This ensures that statistical significance is not overstated due to repeated testing.
>
> Nevertheless, our primary message is that current C-SEO methods are largely ineffective. So we will remove the claim that “LLM Guidance” is promising as suggested by the reviewer.
>
> [1] Holm, S. (1979). “A simple sequentially rejective multiple test procedure”. Scandinavian Journal of Statistics.
>
>
> **Clarifications and Minor Corrections**
>
> Thank you for the additional feedback, below are our answers.
>
> * “t_i is not defined”
>
> $t_i$ is a token. We will define it explicitly.
>
> * “Line 126: Should “C_i” be replaced by “c_i”
>
> “C_i” should indeed be lowercase. We will correct this.
>
> * Missing definition of Tech. Term method.
>
> We apologize for this missing description. This method augments a document with additional technical terms in an attempt to increase credibility. We will add its description to the paper.

---

> > ### Author Response · Authors · 2025-08-05
> >
> > Dear Reviewer 73q6,
> >
> > Thank you for your time reviewing our paper. We hope our rebuttal addressed your concerns. Please let us know if you have any further questions.
> >
> > Thank you.

---

> > > ### Comment · Reviewer_73q6 · 2025-08-07
> > > **Thank you for clarifications**
> > >
> > > Dear authors,
> > >
> > > Thank you for your clarifications. All answers are clear. A got the key thing in the significance thresholds.

---

### Official Review · Reviewer_85kP · 2025-07-05

**Rating:** 4
**Confidence:** 4

**Summary:**

The author proposes a new benchmark for C-SEO methods evaluation, which covers more domains and tasks. The author conducts comprehensive experiments based on the proposed benchmarks, and reveals that existing C-SEO methods shows limitations in influencing the citation rankings. The findings and the proposed benchmarks have the potential to incentivize further research into developing more effective C-SEO methods.

**Dataset Code Accessibility:**

Yes

**Ethical Considerations:**

No, there are no or only very minor ethics concerns

**Final Justification:**

The further experiments and results provided by the author eliminate my concerns and the proposed method is inspired for the area, so i will keep my score and recommend acceptance for the paper.

**Limitations Weaknesses:**

1. The author runs the CSE engine using only the gpt-4o-mini model. Could this introduce bias, reflecting the preferences or limitations of gpt-4o-mini? Does the author have plan for adopting various CSE engine?
2. What is the order of the documents for each query in the constructed dataset? Does the order reflects the ranking of SEO and may influence the effectiveness of C-SEO?

**Strengths Contributions:**

1. The author provides a comprehensive C-SEO benchmarks which covers more domains, tasks, and multiple actors. It has not been explored bofore.
2. The author conducts comprehensive experiments based on the proposed benchmarks, and reveals important limitations current C-SEO methods have.
3. This paper is well written and easy to follow.

---

> ### Author Rebuttal · Authors · 2025-07-29
>
> Thank you for your thoughtful comments. We are glad you found our benchmark novel, our experiments comprehensive, the identified limitations of current C-SEO methods important, and the paper clear and well-written.
>
>
> **Answers to the weaknesses**
>
> 1. “CSE engine only with gpt-4o-mini. Could it introduce bias?”
>
> Certainly possible, but we haven’t seen any evidence yet. In fact, we also ran the conversational search engine with Claude 3.5 Haiku (line 254). The results, shown in Table 4 of Appendix D, lead to the same conclusion: existing C-SEO methods are largely ineffective. To make this clearer, we will move the Claude results into the main paper. We have also conducted new experiments with reasoning models (OpenAI o3 and o4-mini), and again, the conclusions are unchanged. We include these new tables at the end of this rebuttal and will add them to the revised manuscript.
>
> 2. “What is the order of the documents for each query? Can it influence the effectiveness of C-SEO?”
>
> This is an important consideration, and we took it into account in our experimental design. The document order is randomised in all our experiments to control confounders. In particular, the document that applies a C-SEO method can be in any position in the LLM context, so in this way, the ranking from SEO does not impact the evaluation of C-SEO.
>
> **New tables for response 1**
>
> New results using OpenAI o3 as Conversational Search Engine.
> | **Method**         | **retail**        | **videogames**    | **books**         | **web**           | **news**          | **debate**        |
> |--------------------|-------------------|-------------------|-------------------|-------------------|-------------------|-------------------|
> | Authoritative      | -0.04 +/- 1.30    | 0.08 +/- 1.04     | 0.08 +/- 0.95     | 0.01 +/- 0.60     | -0.03 +/- 0.67    | -0.05 +/- 1.36    |
> | Statistics         | -0.33 +/- 1.30    | 0.06 +/- 0.99     | -0.17 +/- 1.04    | -0.25 +/- 0.75    | -0.11 +/- 0.57    | -0.78 +/- 1.78    |
> | Citations          | -0.08 +/- 1.11    | 0.07 +/- 1.08     | -0.01 +/- 0.90    | 0.05 +/- 0.65     | -0.06 +/- 0.69    | 0.11 +/- 1.31     |
> | Fluency            | 0.05 +/- 1.26     | 0.10 +/- 1.01     | 0.02 +/- 0.93     | 0.00 +/- 0.56     | 0.00 +/- 0.57     | 0.13 +/- 1.33     |
> | Unique Words        | -0.05 +/- 1.17    | **0.14 +/- 1.09** | -0.04 +/- 0.97    | -0.02 +/- 0.65    | -0.05 +/- 0.69    | 0.05 +/- 1.54     |
> | Technical Terms     | 0.06 +/- 1.16     | 0.06 +/- 1.04     | 0.01 +/- 1.02     | 0.04 +/- 0.62     | 0.01 +/- 0.67     | 0.08 +/- 1.49     |
> | Simple Language     | -0.01 +/- 1.17    | 0.06 +/- 1.09     | -0.06 +/- 1.10    | 0.02 +/- 0.68     | -0.00 +/- 0.70    | 0.03 +/- 1.30     |
> | Quotes             | -0.11 +/- 1.19    | 0.08 +/- 1.09     | -0.04 +/- 1.03    | -0.01 +/- 0.66    | -0.04 +/- 0.70    | 0.12 +/- 1.49     |
> | LLM Guid.          | 0.06 +/- 1.28     | **0.21 +/- 1.11** | -0.11 +/- 1.15    | 0.02 +/- 0.69     | -0.10 +/- 0.76    | 0.03 +/- 1.47     |
> | Content Improvement | 0.05 +/- 1.27     | **0.14 +/- 1.09** | 0.02 +/- 1.00     | -0.06 +/- 0.75    | -0.09 +/- 0.73    | -0.02 +/- 1.33    |
> | Best SEO           | **2.15 +/- 2.12** | **1.12 +/- 1.68** | **1.41 +/- 1.83** | **0.48 +/- 1.02** | **0.38 +/- 1.01** | **1.36 +/- 1.75** |
>
>
>
> New results using OpenAI o4-mini as Conversational Search Engine
> | **Method**         | **retail**        | **videogames**    | **books**         | **web**           | **news**          | debate            |
> |--------------------|-------------------|-------------------|-------------------|-------------------|-------------------|-------------------|
> | Authoritative      | 0.03 +/- 0.97     | -0.01 +/- 1.01    | -0.10 +/- 0.99    | -0.04 +/- 0.64    | -0.02 +/- 0.40    | -0.21 +/- 1.45    |
> | Statistics         | -0.43 +/- 1.15    | 0.00 +/- 0.97     | -0.55 +/- 1.35    | -0.27 +/- 0.70    | -0.08 +/- 0.41    | -1.15 +/- 1.60    |
> | Citations          | 0.02 +/- 1.14     | -0.05 +/- 1.05    | -0.11 +/- 0.98    | -0.02 +/- 0.53    | -0.06 +/- 0.43    | -0.35 +/- 1.57    |
> | Fluency            | 0.00 +/- 1.09     | -0.01 +/- 1.07    | -0.03 +/- 1.00    | -0.02 +/- 0.56    | -0.03 +/- 0.44    | -0.11 +/- 1.40    |
> | Unique Words        | -0.01 +/- 1.03    | -0.03 +/- 1.13    | -0.09 +/- 0.99    | -0.06 +/- 0.49    | -0.07 +/- 0.45    | -0.25 +/- 1.53    |
> | Technical Terms     | 0.09 +/- 1.10     | 0.00 +/- 1.13     | -0.09 +/- 0.97    | -0.06 +/- 0.66    | -0.01 +/- 0.43    | -0.26 +/- 1.29    |
> | Simple Language     | 0.01 +/- 1.11     | -0.04 +/- 1.19    | -0.06 +/- 1.09    | -0.02 +/- 0.63    | -0.00 +/- 0.46    | -0.26 +/- 1.53    |
> | Quotes             | -0.05 +/- 1.06    | -0.07 +/- 1.06    | -0.14 +/- 1.01    | -0.11 +/- 0.60    | -0.05 +/- 0.50    | -0.30 +/- 1.50    |
> | LLM Guid.          | 0.07 +/- 1.13     | -0.04 +/- 0.99    | -0.20 +/- 1.15    | -0.07 +/- 0.58    | -0.06 +/- 0.55    | -0.25 +/- 1.62    |
> | Content Improvement | -0.02 +/- 1.06    | -0.04 +/- 1.08    | -0.03 +/- 1.07    | -0.14 +/- 0.62    | -0.11 +/- 0.51    | -0.20 +/- 1.48    |
> | Best SEO           | **1.82 +/- 1.92** | **1.10 +/- 1.92** | **0.66 +/- 1.43** | **0.28 +/- 0.82** | **0.28 +/- 0.73** | **1.30 +/- 1.97** |

---

> > ### Author Response · Authors · 2025-08-05
> >
> > Dear Reviewer 85kP,
> >
> > Thank you for your time reviewing our paper. We hope our rebuttal addressed your concerns. Please let us know if you have any further questions.
> >
> > Thank you.

---

> > ### Comment · Reviewer_85kP · 2025-08-06
> >
> > Thank the author for the thorough reply. It addressed my concerns, and the proposed method reveals the potential limitation of current C-SEO method which may inspire further research. So I recommend acceptance.

---

### Official Review · Reviewer_VfSc · 2025-07-23

**Rating:** 5
**Confidence:** 3

**Summary:**

The paper introduces a dataset to measure how much improvement the importance of a given document is in the final generated response. This is an interesting and very useful dataset. Concretely, the dataset consists of a bunch of domains. For each domain, there is a bunch of queries in the dataset. For each domain and query, the dataset specifies a list of relevant documents. The authors construct this dataset in various domains based on the raw data released in those domains. The authors also show the usefulness of the dataset to do SEO. They consider several baseline methods to modify the content of the documents and measure both unilaterla deviation -- where only one document is modified and the rest are unchanged, as well as joint or multiple documents making a deviation. An interesting observation they see is that as the fraction of players/documents that are modified, the delta/improvement to any one document is lowered -- reflecting the zero-sum nature of the metric.

**Dataset Code Accessibility:**

Yes

**Ethical Considerations:**

No, there are no or only very minor ethics concerns

**Limitations Weaknesses:**

The authors were quite transperent in their limitations. For instance, the authors did not investigate if their dataset can help identify the interplay between SEO and C-SEO methods more deeply. The other shortcoming is also that the size of the document/dataset is relatively small. Thus, every document can be ingested into LLM's context for generating the report wuth citations. Presumably in cases where the documents are large so that the final report generator cannot consume it all in its context window may bring about additional levers for players to manipulate in C-SEO. Nevertheless, the paper is well written and executed and makes a clear improvement over state of art.

**Strengths Contributions:**

The paper provides a clear way to evaluate C-SEO methods. It draws the dataset from established sources in application domains such as retail and news. This is important as it is not synthetically generated from LLMs and makes the queries and domain adherence quite useful for applications to test. They also demonstrate that their benchmarks exhibit expected properties such as diminishing returns for any given document if many players/document adapt the strategy to manipulate the content. The paper also teases out the effect of search-ranking (or retrieval) aka classical SEO methods from the content-modification methods using LLMs. That experiment is interesting and valuable to build off of in future works.

---

> ### Author Rebuttal · Authors · 2025-07-29
>
> Thank you for your insightful comments. We are excited that you find the creation of our benchmark sound and useful and that our experiments are interesting and valuable.
>
> **Answers to the weaknesses**
>
> 1. “Did not investigate […] the interplay between C-SEO and SEO”
>
> We agree that the analysis of how applying C-SEO can affect SEO performance is an interesting topic. We believe that this analysis requires a different experimental setup focused on the retrieval stage (i.e., how C-SEO affects SEO rankings) rather than on the ordering of citations within generated answers, which is the focus of our current study. We consider this a promising but separate line of work, which we leave for future research.
>
> 2. “Documents fit the LLM context.” Real-life documents could be so large they don’t fit.
>
> We agree that in real-world settings, some documents may not fit the LLM’s context window. In such cases, summarization can be a viable solution. Our “LLM Guidance” method covers this case by creating summaries of the documents in markdown format. Given the context size of current commercial LLMs, our data fortunately fit the context. Extending our work to scenarios where only partial documents (or summaries) fit the context is a valuable direction. We will add it in our future work section.

---

> > ### Author Response · Authors · 2025-08-05
> >
> > Dear Reviewer VfSc,
> >
> > Thank you for your time reviewing our paper. We hope our rebuttal addressed your concerns. Please let us know if you have any further questions.
> >
> > Thank you.

---

### Author Response · Authors · 2025-08-08

Dear Reviewers,

Thank you for your constructive reviews. The author response discussion period ends today, August 8th. While we have received replies from most reviewers, we would appreciate any final feedback from the remaining reviewers if time allows. To aid in closing the discussion, we would like to summarize the main points raised in the reviews and our responses:

**Strengths**
* **Clear Motivation & Relevance**  (Reviewers 73q6, wqWW)
    * The benchmark addresses a timely and important issue: can documents be enhanced to improve their citation ranking in conversational search engines?
* **Comprehensive and Realistic Benchmark**  (Reviewers VfSc, 85kP, wqWW)
    * The benchmark spans multiple **domains** (retail, news, books, games, web questions, debates) and **multi-actor settings**, reflecting realistic use cases.
* **Insightful Findings on C-SEO vs SEO**  (Reviewers VfSc, 85kP, wqWW)
    * The paper demonstrates that **traditional SEO remains dominant**, and current C-SEO methods are largely ineffective, with diminishing returns as more documents adopt them.
* **Well-Written and Easy-to-Follow Paper**  (Reviewers VfSc, 85kP, wqWW)
* **Important Contribution to the Field** (Reviewers VfSc, 73q6, wqWW)
    * The paper **challenges assumptions** in the community and provides a strong foundation for future work.

**Weaknesses and Our Responses**
* **Interplay Between C-SEO and SEO Not Studied** (VfSc)
    * We **included it in our limitations section** and left it as interesting **future work** due to the **differences to our current setup**.
* **Documents Are Small Enough to Fit LLM Context** (VfSc)
    * True for our current data. In real-world cases, summarization would be needed. **Our experiments with the “LLM Guidance” method already explore this.**
* **Potential Bias from Using Only GPT-4o-mini** (85kP)
    * The paper also **includes** evaluations on **Claude 3.5 Haiku** and we provided experiments in the rebuttal  on **OpenAI o3 and o4-mini**. They all confirm the same findings. **The reviewer responded and agreed with our answer.**
* **Document Order Might Affect Evaluation** (85kP)
    * All documents are **randomly ordered** to avoid position bias. **The reviewer responded and agreed with our answer.**
* **Only Modest Dataset Scale Improvement** (73q6)
    * The **key contribution is not scale but the challenge of a core assumption**: that C-SEO methods are effective. A 10x increase with diverse domains was sufficient to show that current C-SEO methods are not effective. **The reviewer responded and agreed with our answer**.
* **Concerns about Statistical Significance of LLM Guidance method (p-values)** (73q6)
    * We controlled for multiple comparisons via the **Bonferroni-Holm correction** (Table 3 caption) and hence the **statistical significance is not overstated**. Nonetheless, we will remove claims of superiority for the LLM Guidance method as it is not our main claim and goal. **The reviewer responded and agreed with our answer.**
* **C-SEO Methods Weaker Than SEO** (wqWW)
    * **This is our main finding, not a weakness**. The paper demonstrates that current C-SEO methods fail to outperform traditional SEO, challenging assumptions in the field. **The reviewer responded and agreed with our answer.**
* **Concern About Table 3 Robustness** (wqWW)
    * Table 3 shows means, standard deviations, and p-values (adjusted with Bonferroni-Holm). All p-values are reported in the appendix for transparency. Findings are consistent across multiple models (GPT-4o, Claude 3.5 Haiku, OpenAI o3 and o4-mini). **The reviewer responded and agreed with our answer.**
* **Unclear Suitability of Benchmark** (wqWW)
    * Our benchmark is the first comprehensive and realistic benchmark for C-SEO methods covering multiple domains, tasks, and multi-actor scenarios. These design choices reflect actual C-SEO deployment settings. The conclusion of the work is indeed negative: existing C-SEO methods are not effective. **The reviewer responded and agreed with our answer.**

---

### Decision · Program_Chairs · 2025-09-18

**Decision:**

Accept (poster)

**Comment:**

This paper introduces a benchmark to evaluate conversational search engine optimization methods in various domains and tasks. Reviewers unanimously think the work timely and well motivated, particularly given the popularity of LLM-based search engines. Specifically, reviewers think the dataset is comprehensive and that experiments demonstrate expected properties with insightful findings. All reviewers recommend acceptance, with 2xAccept and 2xborderline accept. AC recommends accept.

===== FINAL UPDATE FROM DB Track PCs ====

The final decision for this paper has been taken by the program chairs after consultation with the SACs. All Senior Area Chairs have ranked papers according to the feedback from the AC during the review process. We decided to leave the original meta-review to reflect the opinion of the AC in light of the initial discussions with reviewers and SAC.